# The association of lipid transfer protein VPS13A with endosomes is mediated by sorting nexin SNX5

Alba Tornero-Écija, Antonia Zapata-del-Baño, Laura Antón-Esteban, Olivier Vincent , Ricardo Escalante

Human VPS13 proteins are implicated in severe neurological diseases. These proteins play an important role in lipid transport at membrane contact sites between different organelles. Identification of adaptors that regulate the subcellular localization of these proteins at specific membrane contact sites is essential to understand their function and role in disease. We have identified the sorting nexin SNX5 as an interactor of VPS13A that mediates its association with endosomal subdomains. As for the yeast sorting nexin and Vps13 endosomal adaptor Ypt35, this association involves the VPS13 adaptor-binding (VAB) domain in VPS13A and a PxP motif in SNX5. Notably, this interaction is impaired by mutation of a conserved asparagine residue in the VAB domain, which is also required for Vps13-adaptor binding in yeast and is pathogenic in VPS13D. VPS13A fragments containing the VAB domain co-localize with SNX5, whereas the more C-terminal part of VPS13A directs its localization to the mitochondria. Overall, our results suggest that a fraction of VPS13A localizes to junctions between the endoplasmic reticulum, mitochondria, and SNX5-containing endosomes.

## Introduction

Cellular homeostasis requires the correct traffic of lipids from their main source of synthesis, the endoplasmic reticulum, to their final destination in the membrane of the different organelles. In recent years, the importance of lipid trafficking through lipid transfer proteins at membrane contact sites (MCSs) has gained further interest with the discovery that VPS13 proteins, implicated in several rare neurological diseases, constitute a new family of lipid transporters that function at various MCSs between different organelles (Scorrano et al, 2019). The large size of these proteins (>350 kD) allows them to fit into the intermembrane gap of MCSs, presumably functioning as a bridge where lipids flow through a large hydrophobic cavity that runs their entire length (Kumar et al, 2018; Li et al, 2020; Dziurdzik & Conibear, 2021; Leonzino et al, 2021).

Yeast cells synthesize a unique Vps13 protein that was initially identified for its function in sorting of carboxypeptidase Y into the vacuole (Bankaitis et al, 1986). Other functions include sporulation, Golgi-endosome trafficking, homotypic Golgi fusion, and mitochondrial homeostasis (Bankaitis et al, 1986; Park & Neiman, 2012; Park et al, 2016; De et al, 2017). In addition, loss of the ER-mitochondria contact sites ERMES can be bypassed by *vps13* suppressor mutations (Lang et al, 2015). The localization of Vps13 at specific MCSs appears to be a dynamic and highly regulated process, which may account in part for these pleiotropic functions. Yeast Vps13 localizes to the nucleus–vacuole junctions established between the ER and vacuole; to the contact sites between mitochondria and endosome (EMJs); and between vacuole and mitochondria in the so-called vacuole–mitochondria patch (vCLAMP) (Lang et al, 2015; Park et al, 2016). The localization of Vps13 at different MCSs is regulated by its interaction with lipids and with other proteins that function as adaptors for the different organelles. Three adaptors have been identified in yeast: Ypt35, a sorting nexin that recruits Vps13 to endosomal and vacuolar membranes (Bean et al, 2018), Spo71, the adaptor for Vps13 localization to the prospore membrane (Park & Neiman, 2012), and Mcp1, responsible for Vps13 localization to the mitochondria (John Peter et al, 2017). The interaction involves a conserved PxP motif in the adaptor and a conserved domain in Vps13 that is formed by a six-repeat sequence containing invariant asparagine residues in each repeat, known as the Vps13 adaptor-binding (VAB) domain (Bean et al, 2018). Other relevant conserved domains in Vps13 include the chorein domain at the N-terminal end, which is part of the larger structural domain enriched in beta-strands that form an elongated hydrophobic cavity involved in lipid transport (Kumar et al, 2018), the phosphatidylinositol-3-phosphate (PI3P)-binding APT1 domain (Rzepnikowska et al, 2017a) and the ATG_C and pleckstrin homology (PH) domains (Rzepnikowska et al, 2017b). A fragment containing the latter two domains have been shown to bind phosphatidylinositol-3,5-bisphosphate (PI3,5P2) in vitro (De et al, 2017) and this PI3,5P2 binding was attributed to the PH domain (Kolakowski et al, 2021). The PH domain of Vps13 also binds Arf1 GTPase, which recruits it to the Golgi membranes (Kolakowski et al, 2021). Vps13 presumably interacts with endosomal membranes, which are enriched in PI3P and also with the ER, but the domains implicated have not yet been defined.

Mammals synthesize four VPS13 (VPS13A, B, C, and D) proteins with a domain architecture similar to that of yeast Vps13. Mutations

Instituto de Investigaciones Biomédicas Alberto Sols, C.S.I.C./U.A.M., Madrid, Spain

Correspondence: o.vincent@csic.es; r.escalante@csic.es

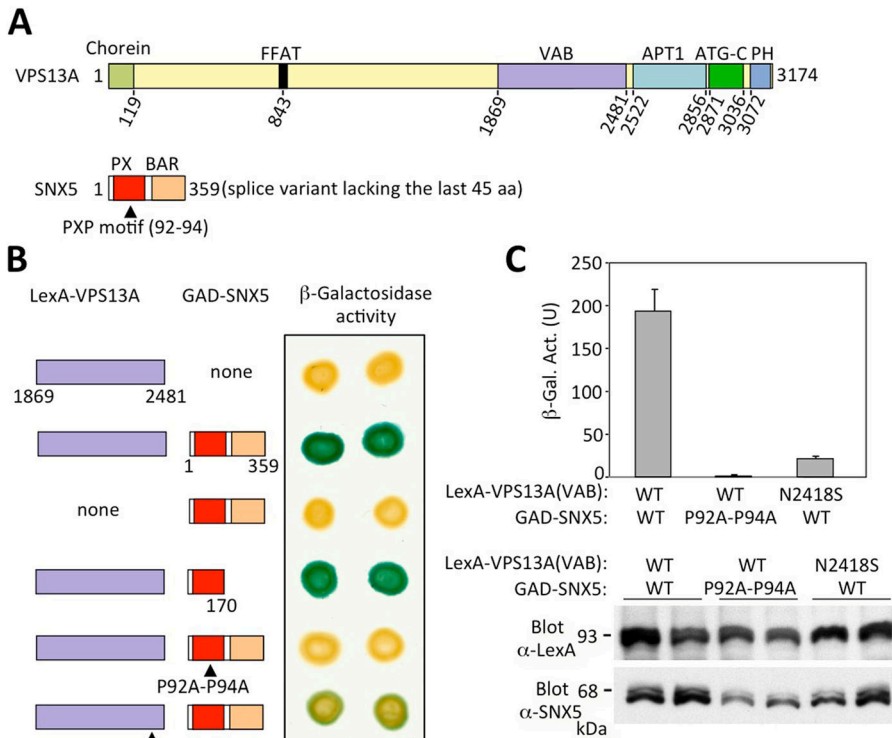

**Figure 1. Interaction of the VPS13A VAB domain with SNX5 in two-hybrid system.**
**(A)** Schematic representation (to scale) of VPS13A and the C-terminally truncated splice variant of SNX5 identified in a two-hybrid screen with the VPS13A VAB domain as bait, showing the positions of the Chorein, VAB, ATG_C, and PH domains in VPS13A and the PXP motif, Phox (PX) and BAR domains in SNX5. Numbers denote amino acid residues. **(B)** Two-hybrid mapping of the VPS13A and SNX5-interacting domains. LexA–VPS13A fusion containing the VAB domain was tested for interaction with GAD-SNX5 or a truncated derivative containing only the Phox domain. The positions of amino acid residue substitutions are shown below. Filter lift assays were developed for 2 h. In each case, eight independent transformants were tested with identical results, of which two are shown. **(B, C)** β-Galactosidase liquid assays were performed to quantify the two-hybrid interaction for the indicated transformants of (B). Wild type forms are denoted as WT. The mean values for four transformants are shown with SD. Below is shown a Western blot analysis of protein extracts of the same transformants with anti-LexA and anti-SNX5 antibodies to visualize the LexA–VPS13A and GAD–SNX5 fusions.

in the corresponding genes cause different neurological diseases: Chorea-acanthocytosis, recently renamed as VPS13A disease (*VPS13A*) (Rampoldi et al, 2001; Ueno et al, 2001; Walker & Danek, 2021), Cohen syndrome (*VPS13B*) (Kolehmainen et al, 2003), Parkinson's disease (*VPS13C*) (Lesage et al, 2016), and spastic ataxia (*VPS13D*) (Gauthier et al, 2018; Seong et al, 2018). The different mammalian VPS13 proteins localize to distinct MCSs (see this rev. [Dziurdzik & Conibear, 2021]). In particular, VPS13A localizes to the ER–mitochondria, ER–lipid droplets, and mitochondria–endosome interfaces (Kumar et al, 2018; Muñoz-Braceras et al, 2019; Yeshaw et al, 2019). VPS13A association with the ER is mediated by the interaction of the ER–protein VAPA with a conserved FFAT motif located in the middle of the rod region (Kumar et al, 2018; Yeshaw et al, 2019). As for the yeast homolog, the most C-terminal region of VPS13A contains several lipid-binding domains such as APT1, ATG_C, and PH (Fig 1A), the latter two being necessary for localization in mitochondria and lipid droplets (Kumar et al, 2018). The PH domain of VPS13A mediates binding to scramblase XK and localization at the interface between the ER and plasma membrane (Guillén-Samander et al, 2022; Park et al, 2022). PH of VPS13A binds yeast Arf1 (Kolakowski et al, 2021), which is very similar to human Arfs, and Arf1 localizes to the mitochondria and Golgi. Although the adaptor-binding VAB domain is conserved in yeast and mammalian VPS13 proteins, the adaptor that binds this domain in VPS13A has not been identified. Identification of mammalian VPS13 protein adaptors is key to understanding the determinants of their localization and function at specific MCSs. However, research in this field has been hampered by the fact that the Vps13 adaptors identified in yeast do not have clear homologs in mammals. Recently, it has been

reported that the interaction of the VAB domain of VPS13D with MIRO is responsible for its localization to the mitochondria and peroxisomes (Guillén-Samander et al, 2021), supporting the idea that the VAB domain could also be a general localization determinant for mammalian VPS13 proteins. Based on this hypothesis, we used the VAB domain of VPS13A in a yeast two-hybrid screen of a human cDNA library, resulting in the identification of the endosomal protein SNX5 as a novel adaptor for the recruitment of VPS13A to SNX5-containing endosome membranes.

## Results and Discussion

### The VPS13A VAB domain interacts with SNX5

To identify potential human adaptors that bind the VAB domain of VPS13A, we performed a two-hybrid screen of a human cDNA library using a LexA DNA-binding domain fusion protein to the VAB domain of VPS13A as bait. We screened $1.5 \times 10^6$ transformants and obtained several positive clones, two of which contained identical plasmids encoding a C-terminal splice variant of the SNX5 sorting nexin lacking the last 45 amino acid residues (Fig 1A). Notably, one of the three yeast Vps13 adaptors is also a sorting nexin, Ypt35 (Bean et al, 2018). The sorting nexins are a diverse family of proteins that contain a Phox domain and are involved in protein sorting and trafficking within the endolysosomal system (Yong et al, 2022). The endosomal protein SNX5 also contains a BAR domain and is part of the retromer complex that regulates the retrograde transport of proteins from endosomes to Golgi (Trousdale & Kim, 2015).

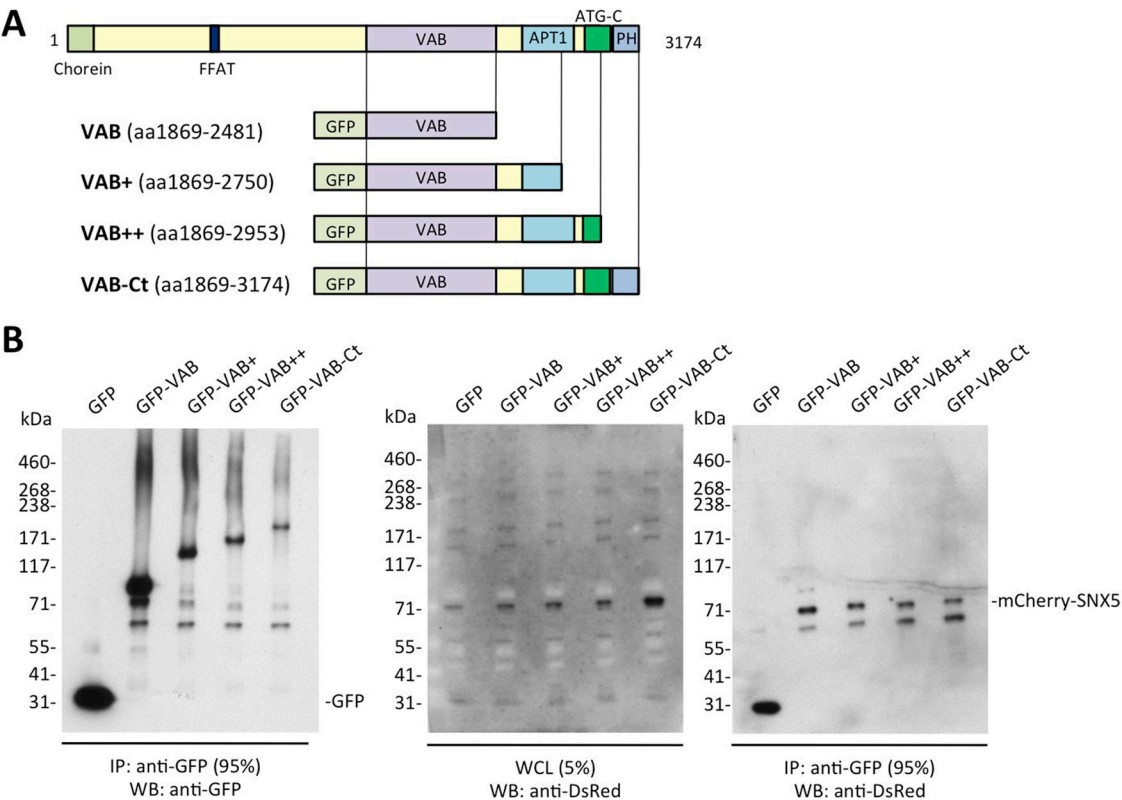

**Figure 2. Interaction of VPS13A VAB domain-containing fragments with SNX5 in HeLa cells.**
**(A)** Schematic showing the VPS13A protein and the generated GFP-fused fragments (VAB, VAB+, VAB++, and VAB-Ct). **(B)** The different GFP-fused fragments were immunoprecipitated from lysates of HeLa cells co-transfected with plasmids encoding these fragments and mCherry-SNX5 encoding plasmid. GFP is included as a negative control. Immunoprecipitates were analyzed by WB (NuPAGE) and the antibodies anti-DsRed recognizing mCherry and anti-GFP antibodies. The percentages of the sample used as WCL (whole cell lysate) and IP (immunoprecipitates) are indicated.

We confirmed the specificity of the two-hybrid interaction in β-galactosidase filter assays and analyzed the determinants of this interaction using truncated and mutant fusion proteins (Fig 1B). Our results indicate that the Phox domain of SNX5 is sufficient to interact with the VAB domain of VPS13A (Fig 1B). Studies in yeast have shown that the binding of Vps13 to adaptors is mediated by repeat 6 in the VAB domain and a consensus PXP motif in the adaptors (Bean et al, 2018; Dziurdzik et al, 2020). In addition, a conserved asparagine residue in repeat 6 of the VAB domain (N2428) plays a key role in this interaction, as alanine substitution of this residue has a strong negative impact on adaptor binding (Dziurdzik et al, 2020). Serine substitution of the respective asparagine residue in VPS13D (N3521S) results in spastic ataxia, suggesting that the molecular mechanisms involved in the interaction between Vps13 and adaptors are evolutionarily conserved. We analyzed the effect of the same substitution (N2418S) in VPS13A and observed a strong reduction of the two-hybrid interaction between the VAB domain and SNX5 (Fig 1B and C). Interestingly, SNX5 contains a PXP motif located in the Phox domain (Fig 1A). Consistent with what was observed in yeast, alanine substitution of the two proline residues of this motif fully prevented interaction between the VAB domain of VPS13A and SNX5 (Fig 1B and C). Taken together, these results support the idea that SNX5 is a novel adaptor of VPS13A and that the molecular basis of the interaction between the VAB domain of

Vps13 proteins and adaptors is conserved in yeast and humans. However, we cannot rule out the possibility that the PXP motif of SNX5 is not directly involved in binding, as it is part of a proline-rich sequence with a possible structural role in the Phox domain (Koharudin et al, 2009). Alternatively, the substitution of proline residues may have an indirect effect on binding through the destabilization of the Phox domain.

Different fragments of *VPS13A* encoding the VAB domain were fused to the GFP gene to validate the interaction of respective protein fragments with SNX5 and to study their subcellular localization in HeLa cells. We used a mCherry fusion to full length SNX5, which includes the C-terminal end missing in the two-hybrid library clone. Protein extracts from HeLa cells co-transfected with plasmids encoding the different VPS13A fragments fused to GFP (Fig 2A) and mCherry-SNX5 were subjected to pulldown. All fragments containing the VAB domain were able to interact with mCherry-SNX5 (Fig 2B), confirming the interaction previously identified in two-hybrid assays in yeast. Sorting nexins are a large family of proteins with up to 33 members in mammalian cells (Hanley & Cooper, 2020). Interestingly, a high-throughput proteomic study has shown that VPS13B coprecipitated with the sorting nexins SNX20 and SNX21 (Huttlin et al, 2015), suggesting that interaction with sorting nexins may be a general property of mammalian VPS13 proteins.

### The VPS13A VAB domain co-localizes with SNX5

Previous studies from our group using the full-length VPS13A bearing an internal GFP tag suggested a major localization in the mitochondria, but also a minor overlapping localization with ER and endosomes (Muñoz-Braceras et al, 2019). We now use different fragments of the VPS13A to more clearly dissect the determinants of its localization and analyze possible colocalization with its interactor SNX5 using confocal microscopy. Interestingly, we found that cells transfected with the constructs encoding the VAB, VAB+, and VAB++ fragments resulted in a punctate pattern over a general cytosolic background, with the smallest fragment, GFP-VAB, diffusing also into the nucleus (Fig 3A). Furthermore, as previously described, additional inclusion of the C-terminal part of the protein containing the ATG_C and PH domains (VAB-Ct) directs the fusion protein to the mitochondria (Kumar et al, 2018) (Fig 3A), although some distinct puncta, often associated to the mitochondria, can also be observed.

The punctate pattern observed with the different fragments of VPS13A (VAB, VAB+, VAB++) would be consistent with localization in endosome-like structures. Strikingly, a complete co-localization was observed when these fragments were produced together with mCherry-SNX5 in HeLa cells (Figs 3B and S1), which further supports the interaction between these two proteins and the endosomal localization of the VAB-containing fragments. Full-length endogenous VPS13A also co-localizes with mCherry-SNX5 in HeLa cells, although only 10% of SNX5 and 10% of VPS13A signal overlap with each other in these cells (Fig S2). This is consistent with previous analysis that revealed a similar level of co-localization of VPS13A with RAB7 (Muñoz-Braceras et al, 2019). These results indicate that the interaction between full-length VPS13A and SNX5 is restricted to a subpopulation of SNX5-containing structures and involves a relatively small fraction of VPS13A. The fact that co-localization is much higher with the isolated VPS13A fragment containing the VAB domain suggests that the other domains of VPS13A that mediate its association with other organelles limit its availability to interact with SNX5.

The importance of the VAB domain defects in pathology is highlighted by a *VPS13D* mutation causing substitution of a conserved asparagine residue in the repeat 6 of the VAB domain, which results in spastic ataxia (Dziurdzik et al, 2020). Notably, introduction of the corresponding mutation into the DNA fragment encoding the VAB+ fragment of VPS13A reduced its level of co-localization with SNX5, in agreement with the results obtained in two-hybrid assays (Fig 3C). These results, together with those obtained for VPS13D (Guillén-Samander et al, 2021), suggest that the VAB domains of mammalian VPS13 proteins act as a critical determinant of the localization of these proteins, as previously reported for the yeast Vps13 protein (Bean et al, 2018).

Although the fragment containing only the VAB domain is able to interact with SNX5 in yeast two-hybrid and pull-down assays, and also co-localize with SNX5 in HeLa cells, we cannot rule out that co-localization of VPS13A and SNX5 in vivo at endogenous levels requires additional determinants, such as the APT1 domain that has been shown to bind PI3P and PI5P (Kolakowski et al, 2020). The synergistic action of both determinants may be critical for the stabilization of the association of VPS13A with specific SNX5-containing endosomes in vivo. This model has been proposed for the yeast-sorting nexin

Ypt35 and the PI3P-binding APT1 domain of Vps13, which together appear to be required for Vps13 recruitment to endosomal and vacuolar membranes (Kolakowski et al, 2020).

### SNX5-VPS13A VAB+ domain puncta localize in endosome subdomains associated with recycling tubular–vesicular structures in proximity to mitochondria and ER

Next, we characterized the nature of SNX5-containing endosomes that co-localize with the VAB domain of VPS13A and the possible MCSs involved. Previous studies in HeLa and HEK293 cells have shown that SNX5 is present in PI3P-enriched domains that are associated (but not completely overlapping) with other endosomal markers, suggesting that SNX5 label-specific endosomal subdomains (Merino-Trigo et al, 2004; Kerr et al, 2006). Despite this endosomal localization, the Phox domain of SNX5 (and that of SNX6 and SNX32) appears unable to interact with PI3P or any other phosphoinositide (Chandra et al, 2019). In contrast, these domains can mediate protein–protein interactions (Paul et al, 2017; Sun et al, 2017; Simonetti et al, 2019). The localization of SNX5 to PI3P enriched-domains could be mediated by its interaction with SNX1 or SNX2 through the BAR domains (Hong, 2019). In addition, SNX5 can also be recruited to the plasma membrane under stimulation of the EGF signaling pathway (Merino-Trigo et al, 2004; Lim et al, 2008). Consistent with these previous data, co-localization analysis with different markers showed that puncta containing SNX5 or the VPS13A VAB+ fragment associate with the early and late endosome markers RAB5 and RAB7, respectively (Fig 4A), although this association is limited to a fraction of VAB+ puncta (17% and 22% of puncta associate with RAB5 and RAB7, respectively; at least 10 cells for each marker were used for quantification). Furthermore, SNX5 puncta were found closely associated with abnormal tubulo-vesicular profiles induced by overexpression of a chimeric form of the cation-independent mannose 6-phosphate receptor (CI-MPR) denoted as CI-MPRtail (Fig 4B) (Waguri et al, 2003). These results are consistent with the described role of SNX5 in retromer-independent retrograde transport of CI-MPR between endosomes and the Golgi (Hara et al, 2008; Kvainickas et al, 2017; Simonetti et al, 2017). The BAR domain of SNX5 and SNX6 is thought to be required to induce the membrane bending necessary for the biogenesis of tubulo-vesicular transport carriers. Notably, our previous studies in VPS13A-depleted HeLa cells have shown defects in cathepsin maturation and reduced lysosomal degradative capacity affecting the efficiency of degradation by autophagy (Muñoz-Braceras et al, 2015, 2019), these phenotypes being compatible with defects in lytic enzymes trafficking to lysosomes. Interestingly, inactivation of *VPS13* in yeast also prevents vacuolar trafficking of carboxypeptidase Y (Bankaitis et al, 1986), most likely because of the defect in endosome-Golgi cycling of Vps10, which plays the same role as CI-MPR in humans (Brickner & Fuller, 1997).

Previous studies described the main localization of VPS13A at the interface between ER and the mitochondria (Kumar et al, 2018; Muñoz-Braceras et al, 2019), although localization between mitochondria and endosomes was also reported by our group (Muñoz-Braceras et al, 2019). To further support the later possibility, we analyzed by video-lapse confocal microscopy the possible association of SNX5 and VAB+-containing vesicles with the mitochondria. Fig 4C and Video 1 and Video 2 show events of co-localization or close

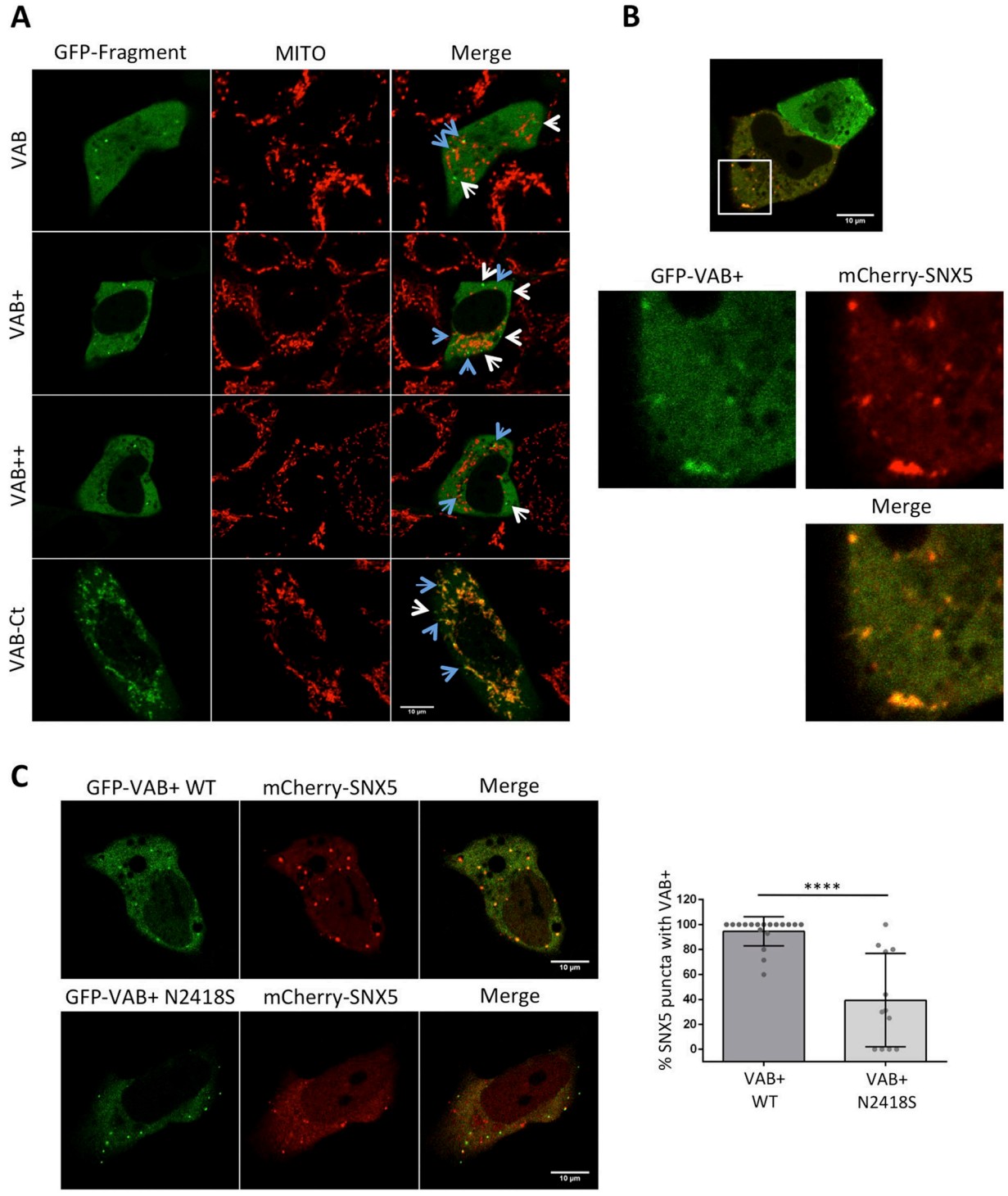

**Figure 3. A VPS13A fragment containing the VAB domain co-localizes with SNX5.**
**(A)** HeLa cells were transfected with the plasmids encoding the different fragments of VPS13A fused to GFP and used for studying the co-localization in vivo under confocal microscopy. Mito Tracker Red was used for co-localization of the indicated fragments with mitochondria (MITO). Examples of puncta that appear associated and non-associated with the mitochondria are marked with blue and white arrows, respectively. **(B)** HeLa cells were co-transfected with the plasmids encoding the VPS13A fragment GFP-VAB+ and mCherry-SNX5 and analyzed using confocal microscopy. Enlarged section of the image is shown. **(C)** HeLa cells were co-transfected with plasmids encoding the VPS13A fragment GFP-VAB+ or GFP-VAB+ (N2418S), and mCherry-SNX5. They were used for studying the co-localization in vivo under confocal microscopy. Analysis of the percentage of mCherry-SNX5 puncta co-localizing with GFF-VAB+ or GFP-VAB+ N2418S fragments is shown. Data correspond to two independent experiments, in which 18 and 12 cells, respectively, were analyzed for each condition. Mann–Whitney test was applied (****$P < 0.0001$).

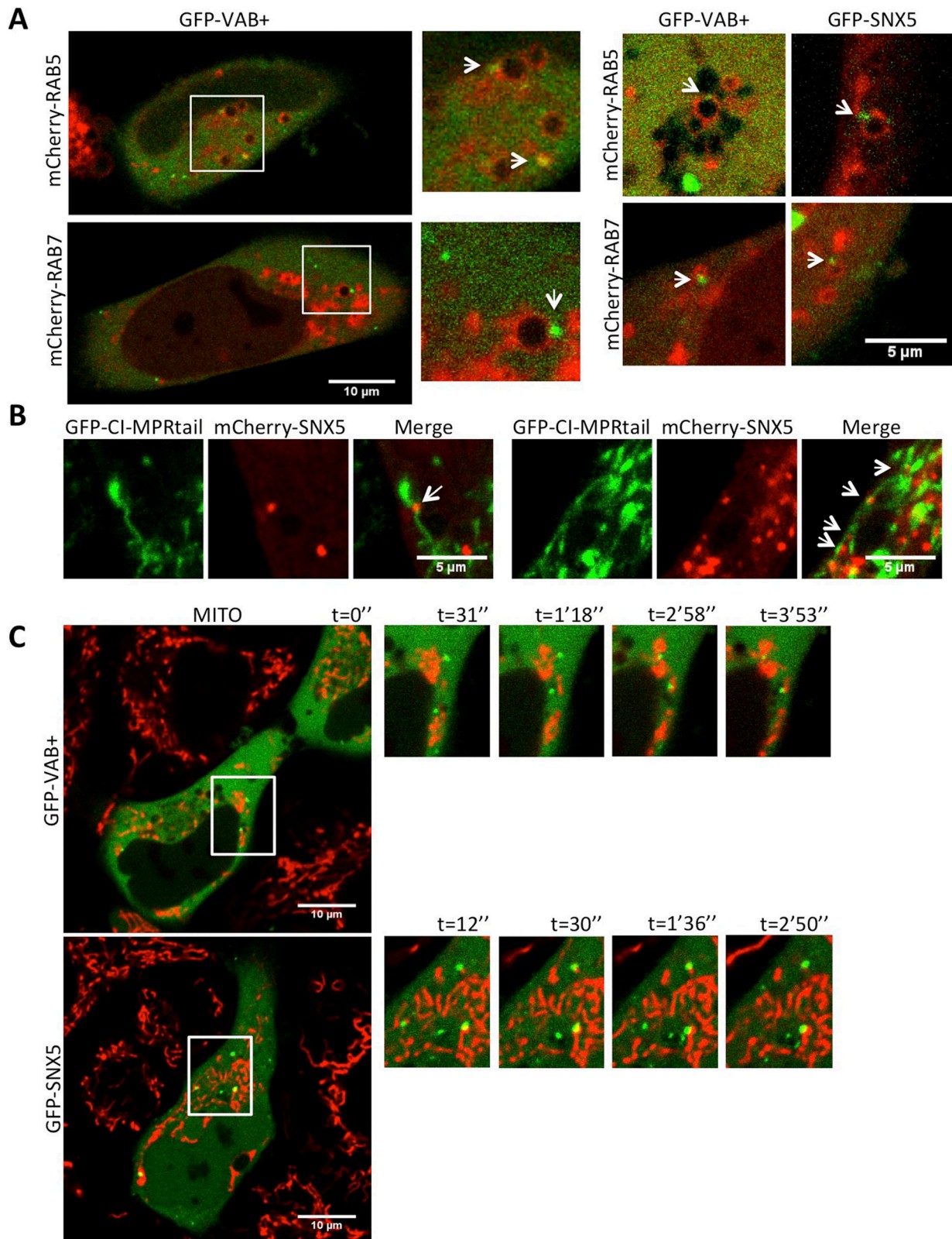

**Figure 4. The VAB domain and SNX5-labeled puncta associate to endosomal structures and mitochondria.**
**(A)** Hela cells were co-transfected with plasmids encoding GFP-VAB+ or GFP-SNX5 for co-localizacion with early endosomes (marker mCherry-RAB5) or late endosomes (marker mCherry-RAB7). Whole cells and enlarged section shown at the right for GFP-VAB+. Additional enlarged sections shown at the left for both GFP-VAB+ and GFP-SNX5. **(B)** HeLa cells were transfected with plasmids encoding GFP-CI-MPRtail and mCherry-SNX5 and were observed in vivo using confocal microscopy. Regions

association overtime between SNX5 or the VAB+ fragment with the mitochondria, indicating a transient tethering of the two organelles.

Recent structural studies suggest that the elongated shape of the VPS13 proteins fits into the gap between organelles and acts as a bridge allowing lipid flow between the two organelles in MCSs. Consistent with this model, the N-terminal region of VPS13A and VPS13C contains an FFAT motif that interacts with the ER VAPA/B proteins, whereas their C-terminal region contains the determinants for interaction with mitochondria (in both VPS13A and C) or endosomes (in VPS13C) (Kumar et al, 2018). Our results now suggest that the C-terminal region of VPS13A can also interact with SNX5-containing endosomes in addition to mitochondria. The VAB domain of VPS13A mediates the interaction with endosomes through SNX5, whereas the ATG_C and PH domains are responsible for the association with mitochondria. Because both determinants are topologically located at the C-terminal end of the protein but at different sites, the possibility of a three-way junction between the ER, mitochondria, and SNX5-containing endosomes is possible. Consistent with this hypothesis, we were able to identify triple co-localization events between the VAB+ fragment or GFP-SNX5, mitochondria, and ER by confocal microscopy (Figs 5A, S3, and S4). The VAB+ fragment cannot be responsible for the establishment of three-way junctions as it lacks the ER and mitochondrial-binding determinants. However, we believe that SNX5 mediates its recruitment to endosomal domains that in some cases are associated with preexisting three-way contacts, probably mediated by additional proteins. Three-way interactions among ER, mitochondria, and lysosomes have been previously documented by cell imaging (Valm et al, 2017), and a four-way contact among ER, mitochondria, Golgi, and lysosomes is required for mitochondrial division (Nagashima et al, 2020), suggesting that the interplay between multiple organelles may be more important that initially thought.

In summary, although previous results indicate a major localization of VPS13A in the ER-mitochondrial junctions (Kumar et al, 2018; Muñoz-Braceras et al, 2019), now our results support the idea that a minor fraction of VPS13A localizes in the interphase between these junctions and a subpopulation of endosomes trough the interaction with the sorting nexin SNX5. This interaction is mediated by the conserved VAB domain in VPS13A and a putative PxP motif in SNX5 (Fig 5B). Although these contacts are minor and transient, the lytic enzymes trafficking defects associated with VPS13A inactivation suggest that this localization could have an important impact on endosome–Golgi retrograde transport. The possibility that this trafficking defect is associated with the human disease chorea-acanthocytosis warrants further investigation.

# Materials and Methods

### Two-hybrid analysis

The two-hybrid screen for LexA-VSP13A(VAB)-interacting proteins was carried out in the *Saccharomyces cerevisiae* strain TAT7

(*MATa, leu2-3,112, trp1-901, his3-Δ200, ade2-101, gal80Δ, LYS2: (lexAop)4-HIS3, ura3:(lexAop)8-lacZ*) co-transformed with pNuLex(U)-VPS13A(VAB) and a human fetal liver cDNA library (Clontech). Selection of His+ transformants was performed in the presence of 5 mM 3-aminotriazole (3-AT) and transformants were subsequently screened for β-galactosidase activity using a filter lift assay. X-gal filter lift assays were performed as described previously (Yang et al, 1992) and developed for 2 h. For β-galactosidase quantitative assays, four independent transformants were grown to logarithmic phase in synthetic dextrose (SD) medium and enzymatic activity was assayed in permeabilized yeast cells and expressed in Miller units (Miller, 1972). Standard genetic methods were followed and yeast cultures were grown in SD medium lacking appropriate supplements to maintain the selection for plasmids (Rose et al, 1990).

### Plasmids

Plasmid pNuLex(U)-VPS13A(VAB) encoding a LexA fusion to the VAB domain of VPS13A was constructed by inserting a PCR fragment containing the VPS13A coding sequence codons 1869-2481 into the BamH1 site of pNuLex(U), a *URA3* derivative of pEG202-NLS obtained by replacing the 3xHA cassette of pWS93 (Song & Carlson, 1998) by a cassette containing the LexA and nuclear localization signal sequence of pEG202-NLS. pGAD-SNX5, the positive clone isolated in the two-hybrid screen, is a pACT2 (Clontech) derivative encoding a C-terminally truncated splice variant of SNX5 lacking the last 45 amino acids. Missense and additional C-terminal truncating mutations were obtained by using mutagenic PCR. pmCherry-SNX5 and pEGFP-SNX5 were generated by cloning the full-length SNX5 coding sequences in the polylinker of pmCherry (Bueno-Arribas et al, 2021) and pEGFP-C3 (*GenBank Accession Number #U57607*). *VPS13A* fragments were obtained by PCR and inserted into pEGF-C3.

### HeLa cells, transfections, immunofluorescence, and confocal microscopy

HeLa cells were a gift from Dr. Alberto Muñoz. The cells were grown in complete DMEM medium (D5648; Sigma-Aldrich), supplemented with 10% FBS (1027-106; Gibco) and 1X penicillin–streptomycin (15140-122; Gibco). DNA transfection was performed using Lipofectamine 2000 (11668019; Invitrogen) according to the manufacturer's instructions. Briefly, Lipofectamine 2000 and DNA were mixed in serum-free Opti-MEM medium (31985062; Life Technologies), after which the DNA–Lipofectamine 2000 complexes were added to the cells. For most plasmids, the medium was changed to DMEM at 6 h after transfection, with the exception of GFP-SNX5 and mCherry-SNX5, for which the medium was not replaced after transfection. The cells were visualized 24 h after transfection, with the exception of *SNX5* expression plasmids, in which case, they were visualized after 16 h. For *SNX5* expression, cells with low expression levels were chosen.

---

corresponding to the mCherry-SNX5 signal co-localizing or closely associated with endosomal tubules are marked by white arrows. **(C)** HeLa cells transfected with GFP-VAB or GFP-SNX5 and treated with MitoTracker Red labeling mitochondria (MITO) Some puncta are stably associated with the mitochondria over time suggesting tethering of both organelles (Video 1 and Video 2; included in supplementary material).

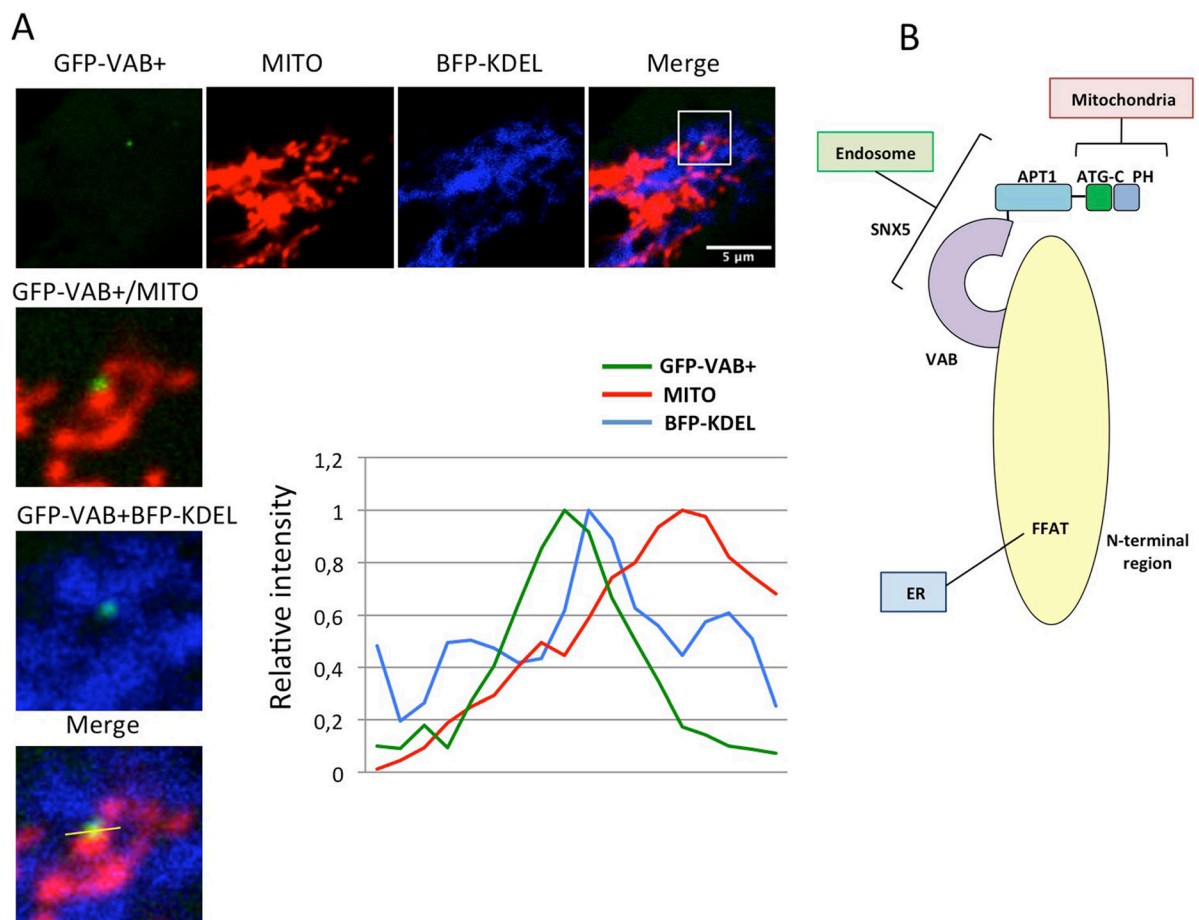

**Figure 5. Possible localization of VPS13A in three-way junctions between ER, mitochondria, and endosomes.**
**(A)** HeLa cells co-transfected with plasmids encoding BFP-KDEL (marker of ER) and GFP-VAB+ were treated with MitoTracker Red labeling mitochondria (MITO), and were analyzed in vivo by confocal analysis to identify possible co-localization between the three markers. The plot represents the relative intensity of the fluorescence signal along the line marked in the enlarged section, showing a partial overlap between the three markers. An additional example is shown in Fig S3. Similar analysis is shown for GFP-SNX5 in Fig S4. **(B)** Schematic model of VPS13A showing the different domains and interactions with organelles (see text for more details).

For in vivo confocal analysis cells were seeded in eight-well plates (μ-Slide eight Well), after which they were transfected with the expression plasmids. 24 h after transfection, the cells were directly visualized using an inverted laser confocal microscope (LSM710; Zeiss), under standard conditions, 37°C and 5% $CO_2$, (Heating System Ibidi). For MitoTracker Red (M7512; Molecular probes) staining, live cells were incubated at 1:5,000 for 15 min.

For immunofluorescence, the cells were seeded on sterile 12 × 12 mm glass coverslips, placed in 24-well plates and transfected as described above. They were fixed with 3% PFA in PBS, pH 7.4, for 30 min at RT. Washes were performed with 1X PBS (133 mM NaCl, 8 mM $Na_2HPO_4$, pH 7.4) and incubated with 100 mM glycine in PBS for 30 min at RT. The cells were permeabilized with cold methanol at −20°C for 10 min. The coverslips were then washed with PBS and incubated with the blocking solution, 3% BSA, 0.2% Triton X-100 in PBS, for 1 h at RT. Subsequently, they were incubated for 3 h at RT with the primary antibodies dissolved in the blocking solution, washed with PBS, and incubated with the appropriate secondary antibodies, also dissolved in the blocking solution, for 1 h at RT. Coverslips were washed with PBS and mounted on a slide using

Prolong Diamond Antifade Mountant (P36970; Molecular probes). Images were acquired using an inverted laser confocal microscope (LSM710; Zeiss) and analyzed with ImageJ software. The anti-VPS13A antibody from Sigma-Aldrich (HPA021662) was used at 1:75.

### Immunoprecipitation and Western blot techniques

For co-immunoprecipitation experiments, the commercial GFP-Trap kit (GFP-Trap A beads; Chromotek, gtak-20) was used following the manufacturer's instructions.

HeLa cells were transfected with the plasmids encoding the different GFP-fused VPS13A fragments, and with the mCherry-SNX5 plasmid. 24 h after transfection, the cells were washed with PBS and suspended in lysis solution (10 mM Tris–HCl, pH 7.5, 150 mM NaCl, 0.5% NP-40), supplemented with protease inhibitors (P8340; Sigma-Aldrich) (1:100) and phosphatases (2.5 mM NaF, 0.2 mM $Na_3VO_4$). Cell lysates were centrifuged at 12,000$g$, 15 min, 4°C and the supernatant was transferred to a new tube. 5% of this supernatant was separated from the sample to obtain total cell lysates, whereas the rest was diluted in washing solution (10 mM Tris–HCl, pH 7.5, 150 mM

NaCl, 2.5 mM NaF, 0.2 mM Na$_3$VO$^4$, and protease inhibitors). The GFP-Trap bead suspension was added to the samples after washing, and the mixtures were incubated for 3 h at 4°C under continuously rotating conditions. Subsequently, the mixtures were centrifuged at 2,000$g$, 2 min, 4°C, after which the supernatant was removed. The GFP-Trap beads were washed five times with the washing solution and the samples were prepared for analysis by WB.

NuPage Tris-acetate 3–8% polyacrylamide gradient gels (EA0378; Invitrogen) were used for the analysis of the co-immunoprecipitation samples. The separated proteins were transferred to PVDF membranes (Immobilon-P polyvinylidene difluoride; IPVH00010; Millipore) by wet transfer. Subsequently, they were blocked in 5% milk powder in TBS-T (50 mM Tris, 150 mM NaCl, 0.5% Tween), for 1 h at RT. The membranes were then incubated with the corresponding primary antibody dissolved in 5% milk powder in TBS-T, at 4°C overnight. The next day, the membranes were washed with TBS-T for 1 h and incubated with the corresponding secondary antibody dissolved in 2% milk powder in TBS-T, at RT for 1 h. Finally, the membranes were washed again with TBS-T for 1 h, before incubation with the chemiluminescent substrate (Super Signal West Pico PLUS Chemiluminescent substrate; 34580; Thermo Fisher Scientific). The chemiluminescent signal was detected using photographic films (CURIX RP2 Plus films; Agfa). The anti-GFP antibody from Sigma-Aldrich (G1544) was used at 1:4,000. The anti-DsRed antibody from Clontech (632496) was used at 1:10,000.

Yeast protein extracts were prepared by the NaOH/trichloroacetic lysis method (Volland et al, 1994) and analyzed by SDS–PAGE and immunoblotting with anti-LexA (R990-25; Invitrogen) or anti-SNX5 (sc-515215; Santa Cruz Biotechnology). The antibodies were detected by enhanced chemiluminescence with ECLPlus reagent (GE Healthcare).

## Supplementary Information

## Acknowledgements

This work has been supported by the "Ministerio de Ciencia, Innovación y Universidades" PGC2018-093604-B-I00 (MCIU/AEI/FEDER, UE) and "Ministerio de Ciencia e Innovación" PID2021-127355OB-I00 (MCIN/AEI/10.13039/501100011033/FEDER, Una manera de hacer Europa). A Tornero-Écija has been supported by the Garantía Juvenil Program from Comunidad de Madrid and Advocacy for Neuroacanthocytosis Patients. We thank Celia Cruz-Cuevas for technical assistance.

### Author Contributions

A Tornero-Écija: formal analysis, investigation, and methodology.
A Zapata-del-Baño: formal analysis, investigation, and methodology.
L Antón-Esteban: formal analysis, investigation, and methodology.
O Vincent: conceptualization, supervision, funding acquisition, investigation, project administration, and writing—review and editing.

R Escalante: conceptualization, supervision, funding acquisition, investigation, writing—original draft, and project administration.

### Conflict of Interest Statement

The authors declare that they have no conflict of interest.

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
