## [Reviewer comments · Life Science Alliance]

The association of lipid transfer protein VPS13A with endosomes is mediated by sorting nexin SNX5

Alba Tornero-Écija, Antonia Zapata-del-Baño, Laura Antón-Esteban, Olivier Vincent and Ricardo Escalante

DOI: 10.26508/lsa.202201852

Corresponding author(s): Dr. Ricardo Escalante (Spanish National Research Council) and Dr. Olivier Vincent

Review timeline:

Submission Date:	2022-11-25
Editorial Decision:	2023-01-03
Revision Received:	2023-03-02
Editorial Decision:	2023-03-15
Revision Received:	2023-03-17
Accepted:	2023-03-17

Scientific Editor: Novella Guidi

Transaction Report:

No Peer Review Process File is available with this article, as the authors have chosen not to make the review process public in this case.

Re: Life Science Alliance manuscript #LSA-2022-01852-T

Dr. Ricardo Escalante
CSIC-UAM
Arturo Duperier 4
Madrid 28029
Spain

Dear Dr. Escalante,

Thank you for submitting your manuscript entitled "The association of VPS13A with endosome subdomains is mediated by SNX5" to Life Science Alliance. The manuscript was assessed by expert reviewers, whose comments are appended to this letter. We invite you to submit a revised manuscript addressing the Reviewer comments.

Thank you for this interesting contribution to Life Science Alliance. We are looking forward to receiving your revised manuscript.

Sincerely,

- A letter addressing the reviewers' comments point by point.
- An editable version of the final text (.DOC or .DOCX) is needed for copyediting (no PDFs).
- High-resolution figure, supplementary figure and video files uploaded as individual

files: See our detailed guidelines for preparing your production-ready images, <https://www.life-science-alliance.org/authors>

B. MANUSCRIPT ORGANIZATION AND FORMATTING:

RE: Life Science Alliance Manuscript #LSA-2022-01852-TR

Dr. Ricardo Escalante
Spanish National Research Council
Arturo Duperier 4
Madrid, Madrid 28029
Spain

Dear Dr. Escalante,

Thank you for submitting your revised manuscript entitled "The association of lipid transfer protein VPS13A with endosomes is mediated by sorting nexin SNX5". We would be happy to publish your paper in Life Science Alliance pending final revisions necessary to meet our formatting guidelines.

- please address all the specific comments raised by Reviewer 1 and attach a point by point response to the reviewer with all the changes performed
- please add ORCID ID for secondary corresponding author-they should have received instructions on how to do so
- please add the Twitter handle of your host institute/organization as well as your own or/and one of the authors in our system

A. FINAL FILES:

B. MANUSCRIPT ORGANIZATION AND FORMATTING:

Sincerely,

3rd Editorial Decision

17 March 2023

RE: Life Science Alliance Manuscript #LSA-2022-01852-TRR

Dr. Ricardo Escalante
Spanish National Research Council
Arturo Duperier 4
Madrid, Madrid 28029
Spain

Dear Dr. Escalante,

Thank you for submitting your Research Article entitled "The association of lipid transfer protein VPS13A with endosomes is mediated by sorting nexin SNX5". It is a pleasure to let you know that your manuscript is now accepted for publication in Life Science Alliance. Congratulations on this interesting work.

DISTRIBUTION OF MATERIALS:

Again, congratulations on a very nice paper. I hope you found the review process to be constructive and are pleased with how the manuscript was handled editorially. We look forward to future exciting submissions from your lab.

Sincerely,
